# Enhancing BCI-Based Emotion Recognition Using an Improved Particle Swarm Optimization for Feature Selection

**DOI:** 10.3390/s20113028

**Published:** 2020-05-27

**Authors:** Zina Li, Lina Qiu, Ruixin Li, Zhipeng He, Jun Xiao, Yan Liang, Fei Wang, Jiahui Pan

**Affiliations:** 1School of Software, South China Normal University, Guangzhou 510631, China; zina@m.scnu.edu.cn (Z.L.); lina.qiu@scnu.edu.cn (L.Q.); ruixinlee@m.scnu.edu.cn (R.L.); zhipenghe@m.scnu.edu.cn (Z.H.); liangyan@m.scnu.edu.cn (Y.L.); fwang@scnu.edu.cn (F.W.); 2School of Electric Power Engineering, South China University of Technology, Guangzhou 510640, China; junxiao@scut.edu.cn

**Keywords:** electroencephalography (EEG), brain-computer interface (BCI), emotion recognition, feature selection, particle swarm optimization (PSO)

## Abstract

Electroencephalogram (EEG) signals have been widely used in emotion recognition. However, the current EEG-based emotion recognition has low accuracy of emotion classification, and its real-time application is limited. In order to address these issues, in this paper, we proposed an improved feature selection algorithm to recognize subjects’ emotion states based on EEG signal, and combined this feature selection method to design an online emotion recognition brain-computer interface (BCI) system. Specifically, first, different dimensional features from the time-domain, frequency domain, and time-frequency domain were extracted. Then, a modified particle swarm optimization (PSO) method with multi-stage linearly-decreasing inertia weight (MLDW) was purposed for feature selection. The MLDW algorithm can be used to easily refine the process of decreasing the inertia weight. Finally, the emotion types were classified by the support vector machine classifier. We extracted different features from the EEG data in the DEAP data set collected by 32 subjects to perform two offline experiments. Our results showed that the average accuracy of four-class emotion recognition reached 76.67%. Compared with the latest benchmark, our proposed MLDW-PSO feature selection improves the accuracy of EEG-based emotion recognition. To further validate the efficiency of the MLDW-PSO feature selection method, we developed an online two-class emotion recognition system evoked by Chinese videos, which achieved good performance for 10 healthy subjects with an average accuracy of 89.5%. The effectiveness of our method was thus demonstrated.

## 1. Introduction

As an advanced function of the human brain, emotion plays an important role in daily life. Emotion recognition has high application value in the fields of commerce, medicine, education, and human-computer interaction, and has become a research area of great interest [1]. In recent years, researchers usually use emotional materials, such as pictures, sounds, and videos, to induce subjects’ emotions, and analyze their physiological signals to obtain the regularity of emotional changes [1]. In the study of emotion recognition, two emotional dimensions of Russell’s Valence-Arousal emotion model are usually used for emotion evaluation [2]. Physiological signals, such as electroencephalography (EEG), electrocardiography (ECG), electromyography (EMG), galvanic skin response (GSR), and respiration rate (RR), are often used to reflect emotional states. The most commonly used is EEG, due to its good temporal resolution as well as acceptable spatial resolution. Furthermore, EEG has been widely used in brain-computer interfaces (BCIs), and the study of EEG-based emotion recognition may provide great value for improving user experience and performance of BCI applications.

At present, there are two major issues for EEG-based emotion recognition that require further investigation. One issue is that the accuracy of the emotion classification is generally low. A critical step in the EEG-based emotion recognition task is to extract features. Several EEG-based features, including Hjorth features [3], fractal dimension features [4], higher order spectra features [5], power spectral density (PSD) features [6], differential entropy (DE) features [7], rational asymmetry (RASM) features [8], and wavelet features [9] have been successfully extracted and applied in emotion recognition. Some researchers have tried to concatenate the above features to extract more information and improve the performance of emotion recognition. However, Samara et al. [10] studied the effect of different feature vectors on the classification accuracy of affective states, and found that combining extracted features to form a feature vector does not necessarily improve the classification accuracy. Another study [11] showed that different subjects may have different sensitivities to different features. These findings are related to what is called the high-dimensionality issue in EEG, because not all of these features carry significant information about emotions. Irrelevant and redundant features increase the feature space, making pattern detection more difficult, and increases the risk of overfitting. It is, therefore, important to identify subject-dependent features that have a significant impact on the performance of the individual emotion recognition.

There are currently many methods available for feature extraction. For example, Arnau-González et al. [12] selected features from the spatial domain and frequency by principal component analysis, and found that naive Bayes, support vector machine (SVM) RBF, and SVM-sigmoid classifiers can significantly improve the accuracy of classification, in which the best accuracy is 67.7% for arousal and 69.6% for valence. Zhong et al. [13] developed a new approach for EEG feature selection, which transferred recursive feature elimination and implemented a linear least square SVM classification, so that the arousal accuracy reached 78.7% and the valence accuracy reached 78.8%. In addition, as a population-based stochastic optimization method, the particle swarm optimization (PSO) method, is becoming more and more popular in the field of feature extraction due to its simple mathematical operations, a small number of control parameters, fast convergence, and easy implementation. Several PSO variants have shown good performance in function optimization and feature selection [14]. For example, Kumar et al. [15] proposed a supervised PSO-based rough set for feature selection of the BCI multiclass motor imagery task, and the outcome outperformed other algorithms using the same dataset. As an important parameter in PSO, the inertia weight is often used to balance the global exploration and local exploitation of the search process. PSO with linearly-decreasing inertia weight (LDW-PSO) [16] was recommended due to its good performance on optimization problems. However, few studies currently use the PSO method for EEG-based emotion recognition.

The other issue is that there are few studies on real-time emotion recognition systems, especially for online BCI systems. Most previous studies have focused on the offline analysis of EEG-based emotion recognition. A few online EEG-based emotion recognition systems have been reported. For example, Sourina et al. [17] developed an online EEG-based emotion recognition system for music therapy, but they did not present the experimental results and system performances. Pan et al. [18] proposed an EEG-based brain-computer interface (BCI) system for emotion recognition to detect two basic emotional states (happiness and sadness), and achieved an average online accuracy of 74.17%. Daly et al. [19] reported an affective BCI system to detect the current affective states of the subjects while listening to music. Using band-power features and support vector machine (SVM) classifier, they achieved a real-time average accuracy of 53.96% for three arousal levels of eight subjects. Nevertheless, the online BCI-based emotion recognition is still in its infancy.

In this study, a real-time EEG-based emotion recognition BCI systems with high accuracy was developed. In order to improve the discrimination ability of the EEG features, we proposed a modified PSO-based feature selection algorithm for emotion recognition. After signal preprocessing, EEG features extracted from the time, frequency, and time-frequency-domains were used to find emotion-relevant features and correlate them with emotional states. We combined features from different scale dimensions and found the optimal features selection using the modified PSO-based method. In particular, a new strategy called multi-stage linearly-decreasing inertia weight (MLDW) was adopted in the PSO method, for the purpose of easily refining the process of decreasing the inertia weight. Furthermore, we used the SVM classifier to recognize emotions based on the emotion model. The above algorithm was verified using EEG data from the DEAP dataset [20]. Our results show that the accuracy of EEG-based emotion recognition can be significantly improved by using our modified MLDW-PSO feature selection, and this modified MLDW-PSO can be used for online emotion recognition. To further demonstrate our proposed method is valid in emotion recognition, we designed an online experiment evoked by videos stimuli. The modified MLDW-PSO algorithm was introduced into the BCI system as a feature selection tool, and then the BCI system can output real-time feedback of online emotion recognition results.

The rest of this paper is organized as follows. Section 2 introduces methods, including data acquisition and stimuli, graphical user interface (GUI) and BCI paradigm, data processing and algorithms, where algorithms, including the feature extraction, feature selection and MLDW-PSO, and the classification. Section 3 presents three experiments and their results, including two offline experiments and one online experiment. Section 4 offers a discussion of the results. Section 5 provides our conclusion.

## 2. Methodology

### 2.1. Data Acquisition and Stimuli

There are currently many emotion recognition datasets available, such as DEAP [20], MAHNOB-HCI [21], and SEED [8]. DEAP is an affective analysis dataset that consists of EEG and peripheral physiological signals recorded from 32 subjects. These physiological signals were recorded while participants watched 40 one-minute-long music videos. After watching the video, participants measured their subjective arousal, and valence on a discrete nine-point scale for each video according to the Self-Assessment Manikin (SAM) [22]. The data file of each participant contains data and labels in the shape of 40 × 40 × 8064 and 40 × 4 arrays, which represent video/trial × channel × data and video/trial × label (valence, arousal, dominance, and liking), respectively. To assess the performance of our emotion recognition approach, two offline experiments were performed on the DEAP dataset. Before the experiments, we have got DEAP data labeled with four types of emotions, i.e., HAHV (Arousal ≥ 5 and Valence ≥ 5), HALV (Arousal ≥ 5 and Valence < 5), LAHV (Arousal < 5 and Valence ≥ 5), and LALV (Arousal < 5 and Valence < 5).

Furthermore, a total of 10 healthy volunteers (H1 to H10, aged 24–35, mean age 26.3 years, two females) from South China Normal University and South China University of Technology were recruited to attend the online BCI experiment. Written informed consent was obtained from all ten healthy subjects prior to the experiment. We used SynAmps^2^ amplifier (Compumedics, Neuroscan, Inc., Abbotsford, Victoria, Australia), and placed 32 electrodes on the subject’s head according to the international 10–20 system. “HEOR” and “HEOL”, and “VEOU” and “VEOL” were used to record an EOG. All electrodes were referenced to the right mastoid and grounded to the forehead. The EEG signals are amplified and digitized at a sampling rate of 250 Hz. Forty Chinese video clips were selected based on the positive and negative emotion models, with 20 clips for each emotion category (positive and negative). In the selection of videos, we first edited 140 comedy and tragedy scenes (clips) from famous Chinese movies or crosstalk shows, each of which was about 30 s. Then, ten volunteers (not the participants in the BCI experiment) were recruited to watch these clips and evaluate their emotions with a level (i.e., not at all, slightly, or extremely) and a keyword (i.e., positive or negative). Forty Chinese video clips scored by all volunteers as extremely positive or negative were selected for BCI online experiments. Notably, the continuous emotional dimension of Self-Assessment Manikin (SAM) used in our study may vary widely between subjects, and the meaning of scales is subjective [23].

### 2.2. Graphical User Interface (GUI) and BCI Paradigm

The GUI and BCI paradigm are illustrated in Figure 1. In the beginning, the participant first had a baseline of 5 s, and there was a prompt sentence in screen and sound to remind the participant that the following video clip to be experienced is positive or negative. All clips appeared in a pseudo-random order. Within 1 s after the prompt, the subject was exposed to the positive or negative video stimuli for 30 s, and the EEG signals were collected and processed simultaneously. With this interface, the positive or negative emotion could be elicited by the stimuli and detected by the proposed algorithm. Then, a feedback of the online emotion recognition result, determined by the BCI algorithm, appeared in the center of the GUI for 5 s. In this study, a smiling/crying cartoon face was presented as feedback, which represents the detection of a positive/negative emotion, respectively. After a rest of 5 s, the next trial will continue until all 40 trials have been completed.

### 2.3. Data Processing and Algorithm 

For our proposed BCI, the emotion recognition included four progressive steps: preprocessing, feature extraction, feature selection, and classification. The overall framework for emotion recognition in this study is illustrated in Figure 2. The methods and algorithms used in this study are described below.

#### 2.3.1. Preprocessing

In this step, we first applied a notch filter to remove the 50 Hz power-line noise. We then adopted a tenth order minimum-phase FIR filter between 0.1 to 70 Hz to filter the raw EEG signal. Then, a time-domain regression method was applied to reduce ocular artifacts. To eliminate the difference in feature scales, the data of each feature was normalized for each participant in the DEAP dataset and the online BCI experiment.

#### 2.3.2. Feature Extraction 

For the online BCI system, a total of 748 emotion-related EEG features were extracted from the EEG signal on three aspects, namely time-domain, frequency domain, and time-frequency domain in each trial. Figure 3 illustrates the data processing process from EEG recording to feature extraction for each subject. It can be seen from Figure 3 that these 748 features generate a feature matrix.

1. Time Domain Features

In the time-domain analysis, we calculated a total of 6 statistical features, including the mean, standard deviation, the mean of the first and second difference absolute values, and the mean of the normalized first and second difference absolute values. The relative equations for calculating the time-domain related features are shown as follows.

Mean: (1)$$\mu_{X}=\frac{1}{N}\sum_{n=1}^{N}X(n),$$
Standard deviation:(2)$$\sigma_{X}=\sqrt{\frac{1}{N}\sum_{n=1}^{N}{\left(X(n)-\mu_{X}\right)}^{2}},$$
Mean of the 1st difference absolute value:(3)$$\delta_{X}=\frac{1}{N-1}\sum_{n=1}^{N-1}\left|X(n+1)-X(n)\right|,$$
Mean of the normalized 1st difference absolute value:(4)$$\bar{\delta_{X}}=\frac{1}{N-1}\sum_{n=1}^{N-1}\left|\bar{X}(n+1)-\bar{X}(n)\right|=\frac{\delta_{X}}{\sigma_{X}},$$
Mean of the 2nd difference absolute value:(5)$$\gamma_{X}=\frac{1}{N-2}\sum_{n=1}^{N-2}\left|X(n+2)-X(n)\right|,$$
Mean of the normalized 2nd difference absolute value:(6)$$\bar{\gamma_{X}}=\frac{1}{N-2}\sum_{n=1}^{N-2}\left|\bar{X}(n+2)-\bar{X}(n)\right|=\frac{\gamma_{X}}{\sigma_{X}},$$
Therefore, the composition vector of statistical features of the signal is: (7)$$FV_{statistical}=[\mu_{X},\sigma_{X},\delta_{X},\bar{\delta_{X}},\gamma_{X},\bar{\delta_{X}}],$$

2. Frequency Domain Features

In the frequency-domain analysis, power features from different frequency bands EEG rhythms are defined in the frequency domain. A total of three power features are calculated in certain frequency bands as follows.

##### Power Spatial Density (PSD) Features

PSD is an effective index of frequency features in response time series, which can describe the distribution of signal frequency changes when random oscillations generate power. The auto-correlation function $\gamma_{t}(k)$ of random signal *x*(*t*) is defined as Equation (8):(8)$$\gamma_{t}(k)=\frac{1}{2\pi }\int_{-\infty }^{+\infty }S_{t}(w)e^{jwk}dw,$$
where E is the expected value and $\bar{x(t+k)}$ denotes the conjugate function of *x*(*t* + *k*). When the auto-correlation function $\gamma_{t}(k)$ satisfies the absolute integrable condition, its Fourier transform and corresponding inverse transform are expressed by Equations (9) and (10), respectively:(9)$$FV_{psd}=S_{t}(w)=F[\gamma_{t}(k)]=\int_{-\infty }^{+\infty }\gamma_{t}(k)e^{-jwk}dk,$$
(10)$$\gamma_{t}(k)=\frac{1}{2\pi }\int_{-\infty }^{+\infty }S_{t}(w)e^{jwk}dw,$$

When *k* = 0, the auto-correlation function $\gamma_{t}(k)$ denotes the power of the signal. The Fourier transform *S_t_*(*w*) represents the power of the signal at the unit frequency, that is, PSD. In this paper, the band powers of four bands are calculated as the features, which are theta (4–8 Hz), alpha (8–14 Hz), beta (14–30 Hz), and gamma (30–50 Hz).

##### Differential Entropy (DE) Features

DE is one of the most important frequency features, which is effective in emotion recognition [7]. DE is defined by Equation (11):(11)$$FV_{DE}=h(X)=-\int_{-\infty }^{\infty }\frac{1}{\sqrt{2\pi \sigma^{2}}}e^{-\frac{{\left(x-\mu \right)}^{2}}{2\sigma^{2}}}\log(\frac{1}{\sqrt{2\pi \sigma^{2}}}e^{-\frac{{\left(x-\mu \right)}^{2}}{2\sigma^{2}}})dx=\frac{1}{2}\log(2\pi e\sigma^{2}),$$
where the time series X follows a Gaussian distribution $N(\mu ,\sigma^{2})$. Similar to PSD, DE is applied to construct features in four frequency bands.

##### Rational Asymmetry (RASM) Features

RASM is an extension of DE [24], which denotes the ratios between DE of several pairs of hemispheric asymmetry electrodes. RASM can be expressed as Equation (12):(12)$$FV_{RASM}=RASM(i)=h(x_{i}^{left})/h(x_{i}^{right}),$$
where *h*(*x*) is defined in Equation (11), and *i* is the pair number. According to the related studies [24] in RASM features, we selected seven channel pairs, i.e. FP1–FP2, F7–F8, F3–F4, T7–T8, C3–C4, P7–P8, P3–P4.

3. Time-Frequency Domain Features

If the signal is unstable, time-frequency methods can bring up additional information by considering dynamical changes.

##### Features based on Wavelet Coefficient

The wavelet transform (WT) is inherited and developed from the short-time Fourier transform (STFT). It overcomes the shortcomings of STFT and becomes an ideal signal time-frequency analysis and processing tool. The details and approximations of the signal can be obtained by WT [25], which are related to high and low frequencies, respectively. The discrete wavelet transforms (DWT) is one of the frequently used transforms in EEG signal processing. It can decompose a sequence of high-pass and low-pass filters until the desired representative frequency band is obtained. In our work, the recorded EEG signal was first down-sampled to 128 Hz, and then five frequency bands between 4 and 128 Hz were obtained from the ‘db4’ details decomposition of the DWT. Specifically, D5 is decomposed into a theta band (4–8 Hz), D4 is decomposed into an alpha band (8–16 Hz), D3 is decomposed into a beta band (16–32 Hz), D2 is decomposed into a gamma band (32–64 Hz), and D1 is decomposed into a high gamma band (64–128 Hz). Finally, the entropy and energy from each frequency band were calculated according to Equations (13) and (14):(13)$$ENT_{j}=-\sum_{k=1}^{N}(D_{j}{\left(k\right)}^{2})\log\left(D_{j}{\left(k\right)}^{2}\right),$$
(14)$$ENG=\sum_{k=1}^{N}{\left(D_{j}(k)\right)}^{2},$$
where *j* = 1, 2, 3, 4, 5 is the level of wavelet decomposition, and k is the number of wavelet coefficients.

Therefore, the composition vector of wavelet features of the signal is:(15)$$FV_{wavelet}=[ENT_{j},ENG_{j}],$$

#### 2.3.3. Feature Selection Based on MLDW-PSO

Feature selection is an essential step before classification. It is mainly used to eliminate the redundant features and select an informative subset. It can generate smaller dimensions of the classification problem without decreasing the classification accuracy. In this study, we developed a modified PSO method that can find the best feature combination from multiple dimensions. In conventional PSO, each individual in the swarm is called a particle [26]. For each particle $P_{i}$, {*i* = 1, 2, …, *k*} there is a position *X_i_* in a dimension search space with a velocity *V_i_* and a memory of personal best position *pbest*. The stored position *gbest* denotes the best particle found so far. In each population, positions of all particles are updated using Equation (16):(16)$$X_{i}(t+1)=X_{i}(t)+V_{i}(t)$$
The speed of particles is updated according to Equation (17):(17)$$V_{i}(t+1)=w\cdot V_{i}(t)+C1\cdot R1()\cdot (pbest-X_{i})+C2\cdot R2()\cdot (gbest-X_{i}),$$
where *w* is the inertia weight, which is usually a constant in the range [0.1,1] in the standard PSO algorithm. The acceleration coefficients *C*1 and *C*2 are set to 2. *R*1 and *R*2 are random numbers uniformly distributed in (0,1).

Some researchers found that PSO with linearly-decreasing inertia weight (LDW-PSO) performs well on many optimization problems [27]. In LDW-PSO, the inertia weight w is given as:(18)$$w=(w_{s}-w_{e})(t_{\max}-t)/t_{\max}+w_{e},$$
where *w_s_* and *w_e_* are the initial and final values for the inertia weight, respectively, *t_max_* and *t* are the maximum number of iterations and the current iteration.

In this work, a modified PSO method with MLDW strategy (MLDW-PSO) based feature selection was proposed to select the most informative weighted features in an effective way. The MLDW-PSO method is summarized in Algorithm 1. First, particles are initialized randomly. A particle in the swarm represents a feature subset, where the dimensionality is the total number of features. Specifically, $X_{i}^{t}$ denotes the *i*-th particle in the swarm at iteration *t* and is represented by *n* numbers of dimensions as $X_{i}^{t}=[x_{i1}^{t},x_{i2}^{t},\ldots ,x_{in}^{t}]$, where $x_{ij}^{t}$ is the position value of *i*-th particle with respect to the *j*-th dimension (*j* = 1, 2, …, *n*). The velocity and the position of each particle in the swarm are also initialized randomly. Next, the fitness evaluation procedure is performed. The error rate of emotion recognition is used in designing a fitness function. In each iteration, after evaluating the fitness of all particles, the algorithm updates the *pbest* and *gbest*, and then updates the position (Equation (16)) and velocity (Equation (17)) of each particle. The position of a particle represents a selected feature subset. A threshold θ is used to determine whether a feature is selected. If the position value is greater than the threshold θ, the corresponding feature is selected. Otherwise, the corresponding feature is not selected. In this study, the threshold *θ* was set to 0.8 based on our experience. Finally, the PSO-based feature selection will stop when a pre-defined stopping criterion (i.e., the maximum number of iterations) is reached. The features corresponding to the current global best particles are determined as the selected features.

The inertia weight play an important role in the trade-off between the global and local exploration capabilities of the particle [27]. To more effectively control the process of decreasing inertia weight, a modified PSO method with MLDW strategy (MLDW-PSO) was proposed [28]. In this MLDW-PSO algorithm, a group of inertia weights was proposed, which can be expressed as:(19)$$w=\left\{ \begin{array}{ll} (w_{s}-w_{m})(t_{1}-t)/t_{1}+w_{m} & \;0\leq t\leq t_{1}\\ w_{m} & {\;t}_{1}<t\leq t_{2}\\ \left(w_{m}-w_{e}\right)(t_{\max}-t)/(t_{\max}-t_{2})+w_{e} & {\;t}_{2}<t\leq t_{\max} \end{array}\right.,$$
where *t_max_* and *t* are the maximum number of iterations and the current iteration. In this study, *t_max_* = 50, *w_s_* = 0.9, *w_e_* = 0.4, and *w_m_*, *t*_1_, and *t*_2_ are the multi-stage parameters. Table 1 lists the multi-stage parameters of the six selected MLDW strategies, called w_1_-w_6_. The curves of these strategies are plotted in Figure 4.
**Algorithm 1:** MLDW-PSO-based feature selectionInput: Dataset, the set of feature and labelOutput: The accuracy of classificationStep1: //Initialize *x* with random position and *v_i_* with random velocity   // Set *w_s_* = 0.9, *w_e_* = 0.4, *w_m_* = 0.5, *MaxDT* = 10, *t*_1_ = 20, *t*_2_ = 30, θ = 0.8.     *X_i_* < -randomPosition();  *V_i_* < -randomVelocity();  Calculate the fitness value with fitness(*X_i_* > θ);  *Gbest* = *X*_1_;   Determine *pbest,*
*gbest* according to fitness(*X_i_* > θ);Step 2: //find *gbest*, the global best particle  for *t* = 1:*t*_1_    for *I* = 1:*K*      *w* = (*w_s_* − *w_m_*)*(*t*_1_ − *t*)/*t*_1_ + *w_m_*;      Update *X_i_*, *V_i_* according to Equations (16) and (17);      Update *pbest*, *gbest* according to fitness(*X_i_* > θ);    end for   end for   for *t*= *t*_1_ + 1:*t*_2_    for *i*= 1:*K*      *w* = *w_m_*;       Update *X_i_*, *V_i_* according to Equations (16) and (17);      Update *pbest*, *gbest* according to fitness(*X_i_* > θ);    end for   end for   for *t*= *t*_2_ + 1:*MaxDT*    for *i* = 1:*K*      *w* = (*w_m_* − *w_e_*)*(*MaxDT* − *t*)/(*MaxDT* − t_2_) + *w_e_*;      Update *X_i_*, *V_i_* according to Equations (16) and (17);      Update *pbest*, *gbest* according to fitness(*X_i_* > θ);    end for   end forStep 3: // Compute best fitness(result)   result=fitness(*gbest* > θ);

#### 2.3.4. Model Training and Classification

The SVM model has been successfully applied to classification in various domains of pattern recognition, including EEG classification in emotion recognition. In our study, the SVM classifier with the linear kernel based on the popular LIBSVM toolbox [29] was used to classify the data, where all parameters were set to default values.

The emotion recognition for each subject is composed of training phase and testing phase. Using the training data from the training phase, an SVM classifier is first obtained based on the different classes of training feature vectors associated with different emotion states. Specifically, the training feature vectors were obtained using our MLDW-PSO-based feature selection method. For the two classes’ emotion recognition, the feature vectors corresponding to the positive and negative clips are labeled +1 and −1, respectively. For the four classes’ emotion recognition, the feature vectors corresponding to the four types of emotions, i.e., HAHV, HALV, LAHV, and LALV, are labeled 1–4, respectively.

The five-fold cross validation is used to calculate the classification accuracy in the experiments. Feature vectors from all channels were first extracted from the three domains and combined. These feature vectors were divided into five samples. New training feature vectors (features’ indexes) were obtained from the four samples using our feature selection method, which is MLDW-PSO. Then an SVM model was obtained based on the category of emotion states and the newly training feature vectors. Finally, the testing feature vectors from the fifth samples with the indexes of the training feature vectors were fed into the trained SVM model to determine the subject’s emotion state.

## 3. Experiment and Result

### 3.1. Experiment I (Offline)

In Experiment I, we determined the optimal strategy for the MLDW-PSO method. We selected 8 of the 32 subjects in the DEAP dataset (i.e., subjects 4, 8, 12, 16, 20, 24, 28, and 32) to perform the offline experiment. Table 1 lists the multi-stage parameters of the six selected MLDW strategies, called w_1_-w_6_. The conventional LDW-PSO is referred to as W_0_-PSO, and the PSO with MLDW is called W_j_-PSO (*j* = 1, 2, …, 6). The five-fold cross validation is used to calculate the classification accuracy for the different PSO-based multi-stage strategies. Table 2 lists the average accuracies of W_0_-PSO and W_j_-PSO for the eight subjects.

It can be seen from Table 2 that W_6_-PSO strategy (marked in bold) achieved the highest average accuracy of PSO at the 10th, 30th or 50th iteration. To further check the feasibility of the W_6_-PSO strategy, we studied the iteration time of the three strategies (i.e., W_0_-PSO, W_2_-PSO, and W_6_-PSO) with the best accuracies. Table 3 presents the average iteration time to achieve the goal classification accuracy using the three algorithms of W_0_-PSO, W_2_-PSO, and W_6_-PSO in the eight subjects. The average iteration time of eight subjects to achieve the best accuracy is less than 10 s. The PSO with W_6_ strategy is a good choice for solving unimodal problems due to its highest accuracy with fast convergence speed. Thus, we used the W_6_ strategy in Experiment II and III.

### 3.2. Experiment II (Offline)

To investigate which features are better in emotion recognition, we computed the five-fold cross-validation accuracy for each feature type (in Figure 4) and the combination of all features without feature selection. The results of 24 subjects in the DEAP dataset (excluding the eight subjects in Experiment I) are listed in Table 4. The average accuracy of the individual statistic feature, PSD features, DE features, RASM features, wavelet features and the combination of all features without feature selection is 43.13% ± 7.76%, 43.44% ± 10.39%, 45.1% ± 8.13%, 45.31 ± 8.61%, 41.25 ± 10.05%, and 42.19 ± 9.90%, respectively. There are no significant differences between each type of the individual feature and the combination of all features without feature selection (all *p*-values > 0.05). Furthermore, the results show that different subjects have different sensitivities to different features. For example, subject s27 obtained better performance using PSD features than the other features, while subject s13 achieved the best accuracy using RASM features among all the features.

Then, we applied the three feature selection algorithms based on the concentration of all 796 features, i.e., relief, standard PSO, and MLDW-PSO. The MATLAB statistics toolbox provides the implementation of the relief algorithm [30]. For the relief algorithm, we set the size of the selection space to 0.5, which means that we selected the top 50% of relevant or important features (374 features in this study). For the standard PSO and MLDW-PSO, a threshold *θ* that used to determine whether a feature is selected or not was set to 0.8. In this condition, the number of features selected by standard PSO or the proposed MLDW-PSO is about 50% of the original number of features. The five-fold cross validation was also used to calculate the classification accuracy of the three feature selection algorithms. The feature selection and SVM training were performed on four folds, and the emotion was recognized in the fifth fold.

The classification results of the feature selection algorithms for the 24 subjects are listed in Table 4. The average accuracy of the Relief, standard PSO, and MLDW-PSO feature selection is 47.60 ± 7.57%, 72.71 ± 5.56%, and 76.67 ± 6.02%, respectively. The best accuracy for each subject is highlighted in bold. The results show that the accuracy obtained using the Relief, standard PSO and MLDW-PSO feature selection methods are significantly higher than the individual features and the combination of all features without feature selection (all *p*-values < 0.05). Furthermore, the MLDW-PSO method achieves higher accuracy than the standard PSO and the relief method for all the subjects (*p*-values < 0.05).

### 3.3. Experiment III (Online)

To validate the effectiveness of our proposed feature selection method, we designed an online emotion recognition experiment. For each participant, there are a total of 40 trials, including 20 training trials and 20 testing trials described in the BCI paradigm above. Before the experiment, each participant was informed about the procedure of the experiment (shown in Figure 1) and was instructed to sit on a comfortable chair approximately 0.5 m away from a 22-in LED monitor. During the experiment, participants were asked to gaze at the screen and stay as still as possible. Participants performed a calibration run of 20 trials to collect training data, and a SVM model was built based on the training data, then was performed on 20 testing trials. In the training or testing trials, each emotion category included 10 trials, and each trial contained one clip. Furthermore, all clips appeared in a pseudo-random order.

Table 5 presents the accuracy of emotion recognition obtained by our proposed MLDW-PSO feature selection method for 10 healthy participants during the online experiment based on our BCI system. The online accuracy of all healthy participants is between 80% and 100%, with an average accuracy of 89.5% ± 5.68%, which is significantly higher than the random level of 50%. These results confirm that the emotions of healthy subjects can be well evoked and recognized by the emotion recognition system.

## 4. Discussion

In order to effectively identify the emotional state based on EEG, this study focuses on the feature selection methods for constructing feature sets. In this study, we performed two offline experiments using the DEAP dataset and designed an online emotion recognition experiment, all of which used MLDW-PSO method for feature selection. In the offline experiments, different dimensional features were first extracted, then feature selection methods were applied to find the best feature combination, and finally the four emotion types were classified by the SVM classifier. Among the three feature selection methods of Relief, standard PSO, and MLDW-PSO, we found that the MLDW-PSO-based feature selection algorithm achieved the highest accuracy for all the subjects in DEAP dataset. To further validate the efficiency of MLDW-PSO-based feature selection, we developed a real-time emotion recognition system to recognize subjects’ positive and negative emotional states while watching video clips. In the online emotion recognition, the MLDW-PSO feature selection method was used to obtain high accuracy for all 10 healthy subjects. All of the results showed that our proposed MLDW-PSO feature selection method improves the performance of EEG-based emotion recognition.

In our offline experiments, the MLDW-PSO feature selection method achieved the highest average accuracy, which may be due to the following three reasons. First, subject-dependent model was applied to analyze affective states. A recent study [11] claimed that subject-dependent emotion recognition usually performs better than subject-independent emotion recognition. Second, the PSO algorithm heuristically searches for the best combination based on the classification accuracy as the fitness function, rather than just roughly combining features. Third, a new group of nonlinear strategies, called MLDW, was proposed to easily refine the process of decreasing the inertia weight. The results suggest that the PSO with w_6_ strategy is a good choice for solving unimodal problems due to its fast convergence speed.

In this study, the subject-dependent model was used to find suitable emotion related to EEG features. The subject-dependent model avoids problems related to variability between subjects, but an emotion classification model must be built for a specific subject. For the subjects in the DEAP dataset, the accuracy of all features without feature selection (in the subject-independent model) did not show differences from each individual feature (in Experiment II). In fact, different subjects have different sensitivities to different emotion-related features. In this respect, subject-dependent feature selection did enhance the performance in emotion recognition. All the accuracy of the three feature selection methods, (in the subject-dependent model) are significantly higher than the accuracy of each individual feature (*p* < 0.05). These results are consistent with those in the literatures [11,31]. Taken together, we can conclude that the subject-dependent model can achieve higher accuracy than the subject-independent model due to the inter-subjects variability.

For Experiment II, the accuracy of the four-class emotion recognition using the MLDW-PSO algorithm reached 76.67%, which is higher than the latest results reported in the review [32]. Previously, Chen et al. [33] proposed a three-stage decision framework for recognizing four emotions of multiple subjects, and found that the classification accuracy for the same four emotions was 70.04%. Gupta et al. [34] studied the channel-specific nature of EEG signals and proposed an effective method based on a flexible analytic wavelet transform to obtain the above four emotions with an emotion recognition accuracy of 71.43%. In addition, Zheng et al. [8] studied stable patterns of EEG over time for emotion recognition using a machine learning approach, and achieved a classification accuracy of 69.67%. By comparison, our method is more effective in EEG-based emotion recognition.

For the real-time emotion recognition system, an average online accuracy of 89.50 ± 5.68% for recognizing two emotion states using the MLDW-PSO algorithm were attained, which is significantly higher than the chance level. Compared to some similar studies, we further found the availability of our emotion recognition system. Liu et al. [35] proposed a real-time movie-induced emotion recognition system for identifying an individual emotional states, and achieved an overall accuracy 86.63 ± 0.27% in recognizing positive from negative emotions. Using stimuli materials similar to [35], the average accuracy of [36] discriminating self-induced positive emotions from negative emotions is 87.20 ± 8.74%. Jatupaiboon et al. [37] classified happy and unhappy emotions using real-time emotion recognition system triggered by pictures and classical music, and achieved an average accuracy of 75.62 ± 10.65%. The classification accuracy of all these studies is lower than the accuracy of our system in identifying two emotional states. 

In order to verify whether our online emotion recognition system can evoke positive and negative emotions, we plotted topographical maps of the average DE features across trials of ten subjects with happy or sad emotional states in four bands (theta, alpha, beta, and gamma bands). Specifically, the features were averaged across all ten healthy subjects and all trials. As shown in Figure 5, the brain neural activity is different when watching negative videos and positive videos. The brain neural activity map of positive emotions show higher power than negative emotions. In the theta and beta bands, the right frontal lobe area and right temporal area were more activated in the positive emotion state than the negative emotion state. In the gamma band, the left occipital lobe and the right temporal lobe regions show higher power in the positive emotion than the negative emotion. These patterns are consistent with those reported in previous studies [37,38]. These findings also proved that our outstanding recognition accuracies were not the result of EMG activities.

There are few factors that may contribute to the high performance of our result. First, the emotion states of subject are easy to evoked, which thanks to the proper selection of stimulus material. Another possible factor is the setting of online feedback, which will not only focus participants’ attention during the trial, but also inspire the participant to adjust their strategies to regulate their emotions to be consistent with the emotions of the stimulating materials. Furthermore, the application of MLDW-PSO feature selection in BCI system is also an important factor. One possible reason is that the trained classifier used MLDW-PSO to capture more valid information from the new feature vector, thereby enlarging its ability to perform pattern recognition. 

## 5. Conclusions

In summary, the MLDW-PSO-based feature selection could enhance the performance of the emotion recognition and be used as an effective method for the real-time BCI application. There are still some open issues in this study that need to be considered in the future studies, such as the optimization of parameter settings, the real-time comparative application of other methods, and classifications of more other emotional states. Furthermore, we will expand BCI-based emotion recognition in a clinical application. The combination of EEG and other bio-signals will also be investigated in the future.

## Figures and Tables

**Figure 1 sensors-20-03028-f001:**
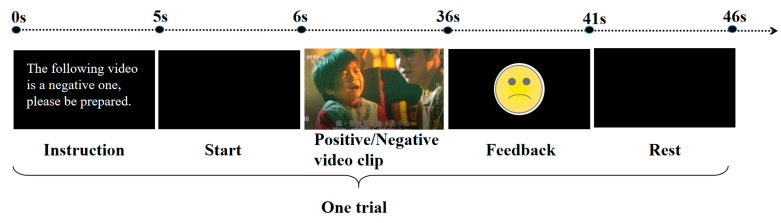
Experimental protocol of our real-time emotion recognition system.

**Figure 2 sensors-20-03028-f002:**
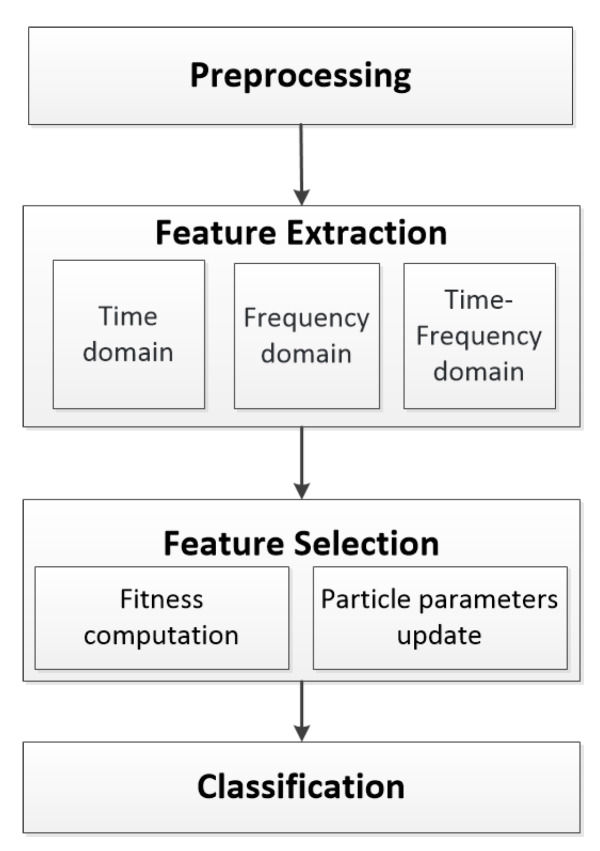
The flowchart of emotion recognition in this study.

**Figure 3 sensors-20-03028-f003:**
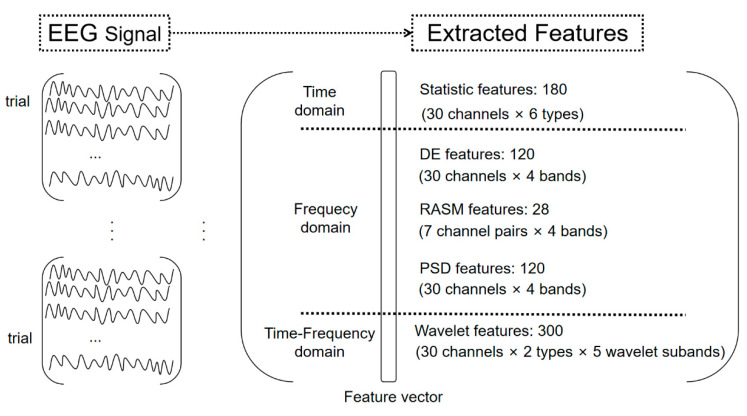
Schematic characterization of data processing from EEG recording to feature extraction in this study.

**Figure 4 sensors-20-03028-f004:**
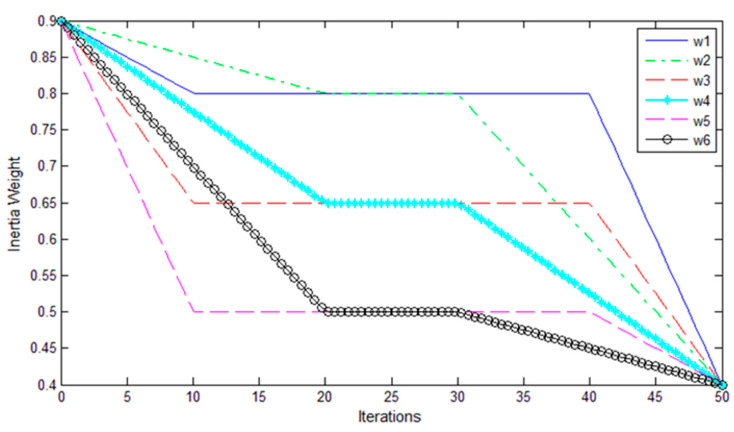
The inertia weight curves over iterations.

**Figure 5 sensors-20-03028-f005:**
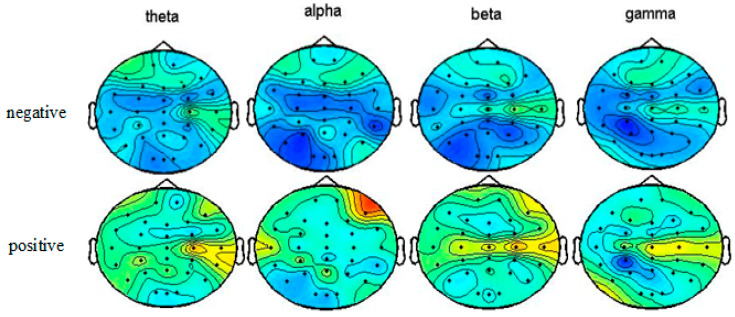
Topographical maps of the average DE features across trials with happy or sad emotional states in the four bands (theta, alpha, beta, and gamma bands) for ten subjects in the online experiment.

**Table 1 sensors-20-03028-t001:** Parameters for selected MLDW strategies.

	w_1_	w_2_	w_3_	w_4_	w_5_	w_6_
w_m_	0.8	0.8	0.65	0.65	0.5	0.5
t_1_	10	20	10	20	10	20
t_2_	40	30	40	30	40	30

**Table 2 sensors-20-03028-t002:** The average accuracy of W_j_-PSO at the 10th, 30th, or 50th iteration for the eight subjects.

Multi-Stage Strategies	10	30	50
W_0_-PSO	78.13	76.56	73.44
W_1_-PSO	75	73.44	75
W_2_-PSO	79.69	78.13	81.25
W_3_-PSO	78.13	73.44	75
W_4_-PSO	70.31	71.88	73.44
W_5_-PSO	75	76.56	73.44
W_6_-PSO	81.25	81.25	82.81

**Table 3 sensors-20-03028-t003:** The iteration times to accuracy using the three algorithms of w0-PSO, w2-PSO, and w6-PSO for the eight subjects.

Subjects	W_0_-PSO	W_2_-PSO	W_6_-PSO	Subjects	W_0_-PSO	W_2_-PSO	W_6_-PSO
s4	2 s	6 s	13 s	s20	13 s	1 s	3 s
s8	2 s	2 s	3 s	s24	2 s	1 s	2 s
s12	2 s	7 s	2 s	s28	2 s	2 s	2 s
s16	5 s	2 s	2 s	s32	6 s	1 s	2 s

**Table 4 sensors-20-03028-t004:** Emotion recognition accuracy based on SVM classifier for 24 subjects with different features vectors. The best accuracy for each subject is highlighted in bold.

Subjects	Statistic Features	PSD Features	DE Features	RASM Features	Wavelet Features	Combination without Feature Selection	Relief Feature Selection	Standard PSO Feature Selection	MLDW-PSO-Based Feature Selection
s1	47.5	25	50	35	35	42.5	40	75	**77.5**
s2	35	55	32.5	42.5	42.5	55	50	70	**82.5**
s3	52.5	47.5	47.5	57.5	62.5	32.5	60	72.5	**77.5**
s5	35	42.5	42.5	42.5	32.5	40	50	62.5	**67.5**
s6	42.5	45	47.5	42.5	45	22.5	45	**72.5**	**72.5**
s7	37.5	32.5	47.5	47.5	37.5	35	50	60	**75**
s9	42.5	42.5	47.5	37.5	37.5	40	50	67.5	**80**
s10	45	37.5	52.5	45	35	32.5	40	67.5	**75**
s11	35	35	35	32.5	22.5	30	37.5	**67.5**	**67.5**
s13	55	50	52.5	65	30	32.5	60	**75**	**75**
s14	57.5	62.5	52.5	52.5	42.5	52.5	40	75	**77.5**
s15	25	45	32.5	35	45	57.5	32.5	75	**77.5**
s17	42.5	40	52.5	42.5	30	47.5	45	**80**	75
s18	37.5	47.5	47.5	47.5	47.5	47.5	52.5	**72.5**	**72.5**
s19	42.5	37.5	42.5	47.5	50	45	50	72.5	**75**
s21	37.5	32.5	37.5	37.5	42.5	25	50	**75**	**75**
s22	30	52.5	45	47.5	52.5	42.5	42.5	**75**	67.5
s23	40	37.5	37.5	50	32.5	47.5	40	70	**82.5**
s25	50	47.5	37.5	32.5	55	50	42.5	**70**	**70**
s26	50	27.5	62.5	37.5	27.5	42.5	47.5	**75**	**75**
s27	57.5	70	55	55	52.5	52.5	62.5	82.5	**85**
s29	50	42.5	47.5	60	42.5	47.5	50	82.5	**85**
s30	37.5	50	30	47.5	55	35	57.5	70	**80**
s31	50	37.5	47.5	47.5	35	57.5	47.5	80	**92.5**
Avg.	43.13	43.44	45.10	45.31	41.25	42.19	47.60	72.71	**76.67**
Std	8.57	10.39	8.13	8.61	10.05	9.90	7.57	5.56	6.02

**Table 5 sensors-20-03028-t005:** The accuracy of emotion recognition obtained by our proposed MLDW-PSO feature selection method for 10 subjects in the online experiment.

Subject	H1	H2	H3	H4	H5	H6	H7	H8	H9	H10	Average Accuracy
Online Accuracy (%)	90	95	95	85	100	85	90	90	80	85	89.50 ± 5.68

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
