# Peer review of "Enhancing BCI-Based Emotion Recognition Using an Improved Particle Swarm Optimization for Feature Selection"

_sensors, 2020, doi:10.3390/s20113028_

Round 1

Reviewer 1 Report

The manuscript proposes the usage of particle swarm algorithm for optimizing (PSO) the set of features for classifications of EEG signals recorded during watching of emotion-laden movies.

The idea in itself is interesting and potentially useful, as clearly described in the Introduction. However, the further parts of this manuscript require clarification before the text could be considered for publication.

My main concern is that the Authors focus on strategies of fine-tuning the PSO algorithm, but the key concept of applying it to the problem of feature selection is not explained.
The critical thing that is not described is:
- How are the features related to the particles?
- What is the individual in the swarm in the context of the feature vector?
- It is entirely not described how the PSO selects the features.
- What is the fitness function used in this application?
- Is there any consistency in the parameters chosen?

Concerning Experiment I:
- How are the selected parameters cross-validated if the features selection and evaluation of the model are performed on the same data set than it is a straight way to overfit.

Concerning Experiment II:
- How many features were selected, and from which groups?
- Do the authors select features and train SVM (or only one of these steps) on four folds and validate them on the fifth fold? - It is not clear.

Concerning Experiment III:
- "In the beginning, the participant first had a baseline of 5 seconds, and there were a prompt sentence in screen and sound to remind the participant whether the following video clip to experience is positive or negative."
-> Does it mean that the participant knew at the beginning of the video what to expect?
Furthermore, in the Discussion: "online feedback, which will not only focus participants' attention during the trial, but also inspire the participant to adjust their strategies to regulate their emotions to be consistent with the emotions of the stimulating materials."
-> The prompt before the video and the feedback may lead to an alternative interpretation of the classified process, namely that the subjects learn to do something that is well classified. Still, it may have no relation to emotions they feel - I could imagine making some forehead frown or other face in case of a negative movie and a calm face for the positive. For a subject, it could be eves subconscious. The EMG signal from such trials would be quite easy to classify. Thus it is not so evident if the presented on-line system is a real-time EEG-based emotion recognition BCI system.

Minor issues:

- "Physiological signals such as electroencephalography (EEG), electrocardiography (ECG), electromyography (EMG), galvanic skin response (GSR), and respiration Rate (RR) are often used to reflect emotional states. The most commonly used is EEG, due to its non-invasiveness, portability, good temporal resolution as well as acceptable spatial resolution"

-> All of these signals are not invasive, and the others, non-EEG, are usually even easier to measure.

- "... amplified and digitized at a sampling rate of 250" -> Add units Hz

- "... to remove the 50 power-line noises." -> Add Hz

- I do not see how the number of features printed in Fig. 3 are obtained. Eg. Time-domain features: there are 6 and number of features is 180 implying there were 30 channels used, but there are 5 PSD bands and 120 features which suggest 24 channels, the number of features of other types also do not fit into some consistent pattern RASM 12 pairs of channels ~ 28 features?

- Also, the arrows in Fig. 3 seem to be misleading, as they suggest that during the preprocessing, some information from different trials mix? Isn't it so that one trial gives one feature vector?

- In PSD feature Authors name 5 bands, but in DE features, they say they evaluate the DE on 4 bands -> Which one is dropped?

- "accelerate speed of particles" -> updated speed?

- Formula 16 -> Empty brackets at R1 and R2, should they depend on t?

- Formula 17, what are w_s w_e and w_e'?

- "First, we applied subject- dependent to analyze affective states." There is something wrong with the grammar.

- In the Discussion section, Authors compare their results to others in the literature but do not take account for the STD. In fact, they present in Table 4 that the acc = 76.48 +/- 6.15, which means that the other cited results are within 1STD, thus not very different.

Author Response

The authors are grateful to the first reviewer for the insightful comments and constructive suggestions. In light of your comments and suggestions, the paper has been revised. Please see our point to point responses in the attached file.

Reviewer 2 Report

In this work, a feature selection algorithm is proposed to recognize subjects’ emotion states based on EEG recordings. Specifically,  a modified particle swarm optimization (PSO) method with multi-stage linearly-decreasing inertia weight (MLDW) is developed for feature selection. The method is used to perform emotion classification on  DEAP data that includes data by 32 subjects. The authors performed two "offline" experiments, one to perform the tuning of their method, one to compare their approach to other algorithms. They also performed an online two-class emotion recognition experiment to test the proposed approach.

The topic has a high scientific interest and it is among the applications of main interest for BCI, but there are some major issues to be discussed:

1. the subjects of the first experiment used for the tuning the parameters are also used in the second experiment and this introduces a bias that results in an overestimation of the performance. The authors should exclude them from the second experiment.

2. In Section 2.3.4 "Model training and classification" it is not clear how the training and testing sets are composed for each subject. The authors should describe these issues in detail.

3. In Section 3.2 they claim:

"Then, we applied the three feature selection algorithms based on the concentration of all features, i.e., Relief, standard PSO, and MLDW-PSO. The MATLAB statistics toolbox provides the implementation of the Relief algorithm. In order to avoid over-fitting and limit the amount of
computation, we set the size of selecting space to 0.5 for these three feature selection algorithms, which means that we selected the top 50% of relevant or important features."

This procedure could compromise the accuracy of the results. In fact, the number of features for the methods should not be fixed in advance. Several methods may achieve higher accuracy with a higher number of features. Therefore, the authors should evaluate each method with an increasing number of features up to the whole set in roder to verify that a method achieves higher accuracy with a given subset of descriptors.

4. It would be more appropriate to carry out the online experiment with different methods beside the proposed one in order to confirm the highest performance.

Some minor points:

a) An example figure with an EEG sample signal showing the features described in section 2.3.2 Feature would increase the clarity of the work.

b) It would be better to unify the mathematical formulation at the beginning of the section 2.3.2.

Author Response

(The authors gave the same response as above.)

Round 2

Reviewer 1 Report

The current version of the manuscript is much improved. However, there are still some issues needing clarification. Below I enumerate them in the order they appeared in the text.

Sect. 2.2
- Did the feedback cartoon inform about detected emotion or congruence/incongruence of the detection and the videos' labeling by the jury?

Sect. 2.3.1
"To eliminate the difference in feature scales, the data of each feature was normalized for each participant in both cases." -> what are the cases?

eq. 9. The integral should be over dk, not dt, and the outer square brackets are not necessary.

"When k=0, the auto-correlation function γt(k) denotes the power of the signal, and its Fourier transform St(w) represents the power of the signal at the unit frequency, that is, PSD."
It sounds like the Fourier transform is computed after substituting k=0, which is, as I think, not what the Authors wanted to say.

"The speed of particles is updated according to formula (16)." -> ...(17)

It is still not very clear: ". A particle in the swarm represents a feature subset, where the dimensionality is the total number of features in the dataset."

I guess what the Authors do is as follows, but it is not clear from the description:
- We have a high dimensional feature space, 796 for DEAP.
- A particle is a vector of weights Pi = (pi1, pi2,....pi796), and the weight pij tells how important a feature j is.
- Then you initialize a set of particles (a swarm), and it evolves according to eq(16) and (17).
- At each step, a fitness function (accuracy) is evaluated for each particle with thresholded vectors Pi, to select updated gbest.
- (what about pbesti ?)
- After reaching the stopping criteria, the best vector Pi is thresholded.

Language, sect. 2.3.4:
"Feature vectors from all channels were first extracted from three domains and combined them. " -> "Feature vectors from all channels were first extracted from the three domains and combined. "?

Language, sect 3.1: "The average iteration time of eight four to achieve the best accuracy is less than 10 s" -> of eight four?

Author Response

The point-to-point responses to the first reviewer’ comments are attached below.

Reviewer 2 Report

The authors have replied exhaustively to the comments of this reviewer.

Author Response

Thank you very much!